# Generational Diet-Induced Obesity Remodels the Omental Adipose Proteome in Female Mice

**DOI:** 10.3390/nu16183086

**Published:** 2024-09-13

**Authors:** Naviya Schuster-Little, Morgan McCabe, Kayla Nenninger, Reihaneh Safavi-Sohi, Rebecca J. Whelan, Tyvette S. Hilliard

**Affiliations:** 1Department of Chemistry, University of Kansas, Lawrence, KS 66045, USA; naviya@ku.edu (N.S.-L.); rwhelan1@ku.edu (R.J.W.); 2Ralph N. Adams Institute for Bioanalytical Chemistry, University of Kansas, Lawrence, KS 66047, USA; 3Department of Chemistry and Biochemistry, University of Notre Dame, Notre Dame, IN 46556, USA; mmccabe7@alumni.nd.edu (M.M.); knenning@alumni.nd.edu (K.N.); rsafavis@nd.edu (R.S.-S.); 4Harper Cancer Research Institute, University of Notre Dame, Notre Dame, IN 46617, USA; 5Department of Chemistry and Biochemistry, Seton Hall University, South Orange, NJ 07079, USA

**Keywords:** diet, adipose tissue, omental tissue, obesity, generational obesity, proteomics, S-Trap, suspension trapping

## Abstract

Obesity, a complex condition that involves genetic, environmental, and behavioral factors, is a non-infectious pandemic that affects over 650 million adults worldwide with a rapidly growing prevalence. A major contributor is the consumption of high-fat diets, an increasingly common feature of modern diets. Maternal obesity results in an increased risk of offspring developing obesity and related health problems; however, the impact of maternal diet on the adipose tissue composition of offspring has not been evaluated. Here, we designed a generational diet-induced obesity study in female C57BL/6 mice that included maternal cohorts and their female offspring fed either a control diet (10% fat) or a high-fat diet (45% fat) and examined the visceral adipose proteome. Solubilizing proteins from adipose tissue is challenging due to the need for high concentrations of detergents; however, the use of a detergent-compatible sample preparation strategy based on suspension trapping (S-Trap) enabled label-free quantitative bottom-up analysis of the adipose proteome. We identified differentially expressed proteins related to lipid metabolism, inflammatory disease, immune response, and cancer, providing valuable molecular-level insight into how maternal obesity impacts the health of offspring. Data are available via ProteomeXchange with the identifier PXD042092.

## 1. Introduction

In 2022, over 890 million adults were considered obese, according to the World Health Organization (WHO) [1]. Despite being a preventable disease, obesity has tripled since 1975 and is now one of the leading causes of death worldwide [1]. The increased prevalence of obesity has led to a significant rise in several diseases, including type 2 diabetes [2], cardiovascular disease [3], sleep apnea [4], and cancer [5]. Most recently, obesity has been associated with increased severity of COVID-19 illness [6,7].

A major contributor to the obesity epidemic is the Western diet, which contains energy-dense foods that are high in fat and sugar [1,7]. In laboratory studies of animal models, animal feed can be controlled to assess the effects of high-fat (Western) and low-fat (control) diets on profiles of protein expression by performing quantitative proteomics analysis on suitable tissue such as adipose [8]. Adipose tissue growth varies according to maternal diet and can contribute to metabolic disorders, such as insulin resistance and impaired glucose homeostasis [9]. While the effect of diet, both low and high fat, on protein expression in adipose tissue has been reported [10,11,12,13,14], the influence of generational or maternal obesity on the remodeling of the visceral omental adipose proteome has not been evaluated.

While adipose tissue is a valuable resource for understanding the molecular-level changes resulting from dietary interventions, it is a challenging matrix for proteomics analysis because of its abundance in lipids. High concentrations of detergent are required to extract and solubilize proteins from adipose [15,16,17]; however, these agents are not compatible with mass spectrometry. To overcome this problem, gel-based approaches are commonly used [16,18]. Alternatively, suspension trapping (S-Traps) is a filter-based method used for proteomics sample preparation that requires high concentrations of SDS [19]. Considering the challenges of working with adipose tissue and the need for strong detergents to solubilize proteins, we hypothesized that S-Traps would be a suitable sample preparation method for adipose tissue. Here, for the first time, we utilize S-Traps to perform proteomic analysis of omental visceral adipose tissue in a female generational obesity cohort study. We have identified unique sets of proteins and protein networks that characterize changes in omental adipose in response to modulation of dam and offspring diet. Overall, the identified differentially expressed proteins were involved in pathways related to cancer, inflammatory disease, and immune response. The greatest difference in proteomic profiles was observed when comparing lean and generational obese offspring, with observed alterations in proteins associated with lipid transport, lipid metabolism, and immune response. These findings illuminate the complex biological mechanisms that regulate adipose tissue and represent a broad-based reference dataset to inform future studies of generational obesity and its impact on peritoneal diseases, including peritonitis, dialysis-associated injuries, and peritoneal metastasis of cancers, such as ovarian, gastric, and pancreatic cancer.

## 2. Materials and Methods

### 2.1. Materials

Sodium dodecyl sulfate (SDS), iodoacetamide (IAA), and triethylammonium bicarbonate (TEAB) were obtained through Millipore Sigma (St. Louis, MO, USA). Tris(2-carboxyethyl)phosphine (TCEP), deoxycholic acid (DCA), phosphoric acid, methanol (Burdick & Jackson, Muskegon, MI, USA), 0.1% formic acid in water (Burdick & Jackson, Muskegon, MI, USA), and phosphate-buffered saline (PBS) were purchased from VWR. Additionally, 99% pure formic acid (FA), acetonitrile (ACN), and C18 ZipTips were purchased from Fisher Scientific (Hanover Park, IL, USA). S-Traps™ were purchased from Protifi (Huntington, NY, USA). Mass spectrometry-grade trypsin gold was ordered from Promega (Madison, WI, USA) and reconstituted according to the manufacturer’s instructions.

### 2.2. Experimental Design and Statistical Rationale

All murine studies were approved by the Institutional Animal Care and Use Committee, University of Notre Dame, and were conducted in accordance with relevant guidelines and regulations of this committee. All experiments were conducted with three biological samples per dietary cohort. Each sample was analyzed in technical duplicate for a total of six LC-MS/MS analyses per dietary cohort. Statistical analysis was carried out using Graph Pad Prism 9.0 software.

### 2.3. Murine Generational Obesity Animal Model

The pre-clinical murine model of diet-induced obesity (DIO) includes two maternal cohorts (dams) of female C57BL/6 mice (Jackson Lab) with intact host immunity fed a control diet (10% fat; control dam (Dam_(CON)_); Research Diets D12450H) compared to animals fed a high-fat Western diet (45% fat, western dam Dam_(WES)_; Research Diets D12451), as illustrated in Figure 1. All male mice (sires) were fed normal breeder’s chow and allowed to mate with a pair of female mice (dams) from each diet cohort. Offspring resulting from both cohorts of dams were allocated to either the control or Western diet (Figure 1). Cohorts are designated based on Maternal Diet_(OFFSPRING DIET)_ as follows: Lean = Con_(CON)_; Transitional Lean = Con_(WES)_; Transitional Obesity = Wes_(CON)_; Generational Obese = Wes_(WES)_. Mice were euthanized, and the omenta were dissected [20].

### 2.4. Protein Isolation and Proteomics Sample Processing

Proteins were isolated from adipose tissue using the Qiagen AllPrep DNA/RNA/Protein Mini Kit (Cat No. 80004, Qiagen, Germantown, MD, USA). Isolated proteins were eluted into nuclease-free water and stored at −80 °C until future use. Before proteolytic digestion, proteins were solubilized by adding SDS and PBS for final concentrations of 5% and 1×, respectively. Total protein was quantified using the Pierce BCA assay.

For each digestion, 20 µg of protein was taken into the reaction. Proteins were denatured and reduced using 7% SDS and 10 mM TCEP at 95 °C for 10 min; 0.2% DCA was included to prevent protein adsorption, and 100 mM TEAB was included as a buffering agent. Following reduction, 5 mM IAA was added, and samples were incubated in the dark for 30 min at RT. The alkylation reaction was quenched using phosphoric acid at a final concentration of 1.2%. The proteins were then prepared for S-Trap digestion following the kit’s instructions. Briefly, 100 mM TEAB in MeOH was added to flocculate proteins. The suspension was spun onto the S-Trap at 4000 g for 30 s. The S-Trap was washed using 100 mM TEAB in water. Then, 1 µg of trypsin in 100 mM TEAB in water was added to the S-Trap. The S-Trap was placed in an incubator set to 37 °C for 4 h. Following digestion, peptides were eluted using 100 mM TEAB, 0.1% FA, 50% ACN, and 0.1% FA. All eluates were collected into the same tube and quenched with formic acid to a final concentration of 1%. All eluates were collected into one vial and dried on a SpeedVac. Peptides were resuspended in 0.1% FA, desalted using ZipTips, evaporated to dryness, and resuspended in water containing 4% ACN and 0.5% FA to 20 µL. Each sample was desalted twice for a total of 10 µg peptide. Each sample was analyzed in technical duplicate.

### 2.5. Liquid Chromatography and Mass Spectrometry Conditions

Peptides were analyzed using a Waters M-Class ultrahigh-pressure liquid chromatography (LC) system coupled to a Q-Exactive High-Frequency (QEHF) mass spectrometer (Thermo Scientific, Waltham, MA, USA). Peptides were eluted from a BEH C18 column (Waters, 100 µm inner diameter × 100 mm outer diameter, 1.7 µm particle size) using a 100 min gradient at a flow rate of 0.9 µL/min. A binary solvent system was used, where Solvent A consisted of water with 0.1% FA, while solvent B consisted of CAN with 0.1% FA (Burdick & Jackson, Muskegon, MI, USA, VWR). The following linear gradient was used for all samples: 4–33% B for 90 min, 33–80% B for 2 min, constant at 80% B for 6 min, and then 80–0% B for 2 min to equilibrate the column. Data were collected in positive ionization mode. Mass spectra were acquired in the Orbitrap at 60,000 resolving power, and tandem mass spectra were then generated for the top seventeen most abundant ions with charge states ranging between 2 and 5. Fragmentation of selected peptide ions was achieved via high-energy collisional dissociation (HCD) at a normalized collision energy of 35 eV in the HCD cell of the QEHF. Samples from three different mice were digested to generate biological triplicates. Each of these digests was analyzed in technical duplicate for a total of six LC-MS/MS analyses per dietary cohort.

### 2.6. Data Analysis

The data analysis was performed using MetaMorpheus version 0.0.320, available at https://github.com/smith-chem-wisc/MetaMorpheus (accessed on 29 September 2022) [21,22]. The mass spectrometry proteomics data have been deposited to the ProteomeXchange Consortium via the PRIDE [23] partner repository with the dataset identifier PXD042092. The following calibrate and search task settings were used: protease = trypsin; maximum missed cleavages = 2; minimum peptide length = 7; maximum peptide length = 45; initiator methionine behavior = Variable; fixed modifications = Carbamidomethyl on C, Carbamidomethyl on U; variable modifications = Oxidation on M, Deamidation on N, Deamidation on Q; max mods per peptide = 2; max modification isoforms = 1024; precursor mass tolerance = ±10.0000 PPM; product mass tolerance = ±0.0200 Absolute; report PSM ambiguity = True. The combined search database contained 55,557 non-decoy protein entries, including 474 contaminant sequences. Label-free quantification (LFQ) was performed using FlashLFQ [24]. A 20 ppm peak tolerance was allowed, and match between runs was enabled. Data files were processed on a computer running Microsoft Windows 10.0.19044 with a 64-bit AMD Ryzen Threadripper PRO 3955WX 16-Cores processor with 32 threads and 128 GB installed RAM. The total time to perform the calibrate and search tasks on 36 spectra files was 395.92 and 218.21 min, respectively.

The LFQ intensity values were used to determine changes in protein abundance, and volcano plots were generated using R-studio (version 2022.2.3.492) [25]. The average and standard deviation of the LFQ intensity values for each of the six injections per cohort were calculated. The log2 fold change was calculated for each comparison group, and *p*-values were calculated using Student’s *t*-test. The significance values were set as fold change >2 or <−2 and a *p*-value less than 0.05. The following R-studio packages were used for data analysis: ggplot2 [26], plotly [27], stringr [28], ggrepel [29], and tidyverse [30]. Venn diagrams were created using Venny 2.1 [31].

Principal component analysis (PCA) was used as an unsupervised method to represent a high-dimensional data structure in a smaller number of dimensions for characterizing objects in the dataset. PCA was applied to quantitative proteomic data using PLS-toolbox version 9.1 [32]. The datasets of all quantified proteins were used to perform a PCA analysis. The data points for each of the groups (i) Con_(CON)_; Con_(WES)_, (ii) Wes_(CON)_; Wes_(WES)_, and (iii) Dam_(CON)_; Dam_(WES)_ were separated into two sets (offspring and dam), and PCA analysis was performed. The first three eigenvalues explained more than 85% of the total variance. The PCA score-plot obtained using the PLS-toolbox, projecting the objects into the subspace, was created by the 1st, 2nd, and 3rd latent variables of the model.

Gene ontology analysis was performed using the PANTHER (Protein ANalysis THrough Evolutionary Relationships) Classification System [33]. UniProt protein accession numbers were used as the input for analysis. Our analyses looked at biological process, cellular component, and molecular function of the genes identified in each comparison group. Pie charts were created in Igor Pro 9.0 (Wavemetrics) using the data from PANTHER.

Differentially expressed proteins (*p* < 0.05) between the dietary groups were subjected to Ingenuity Pathway Analysis (IPA; Qiagen) to determine differential regulation of canonical pathways and functional enrichment analyses.

## 3. Results

### 3.1. Generational Obesity Breeding Scheme and Characterization

A breeding scheme was designed to examine the heritable contribution of maternal high-fat diet as well as the impact of offspring diet on the omental adipose proteome (Figure 1A). Female mice (dams) were exposed to a high-fat diet (Western; Wes) or a low-fat diet (control; Con) for 6 weeks before pregnancy and until weaning of offspring at 4 weeks. A dam from each diet cohort (Dam_(CON)_ and Dam_(WES)_) was bred with a male mouse (sire) and the resulting female offspring from each dam were fed either control or Western diet, resulting in four additional offspring dietary cohorts: Con_(CON)_ (Lean); Con_(WES)_ (transitional obese); Wes_(CON)_ (transitional lean); and Wes_(WES)_ (generational obese) (Figure 1A). As expected, dams fed a high-fat diet had elevated body weight compared to low-fat diet-fed dams (Figure 1B). The first-generation offspring of these dams fed a high-fat diet had an elevated body weight regardless of dam diet (Con_(WES)_ and Wes_(WES)_; Figure 1B). Additionally, the omental weights of high-fat diet-fed offspring (Con_(WES)_ and Wes_(WES)_) were increased when compared to mice only exposed to a control diet (Con_(CON)_; Figure 1C). Adipocyte hypertrophy was observed in generational obese mice (Wes_(WES)_) relative to all the other dietary cohorts (Figure 1D,E). To evaluate the generational dietary influence on omental adipose proteome remodeling, several comparisons were made, as shown in Table 1, with three additional comparisons shown in Appendix A. Although all cohorts and groups are of interest, the primary focus of this report is on the generational obesity and maternal obesity comparison groups.

### 3.2. Proteolytic Digestion of Omental Adipose Tissue Proteins Using S-Traps

Here, we report the first use of S-Traps to digest proteins isolated from omental adipose tissue. The data points for each of the groups (i) Con_(CON)_; Con_(WES)_, (ii) Wes_(CON)_; Wes_(WES)_, and (iii) Dam_(CON)_; Dam_(WES)_ were separated into two sets (offspring and dam), and PCA analysis was performed. Both dam and offspring dietary cohorts were segregated into dietary cohorts by PCA analysis (Figure 2A–D). Using the S-Trap digestion method, we identified over 1000 proteins from most cohorts (Table 2 and Appendix A; Figure 3A,B). In comparison to more traditional methods, including 2D gel electrophoresis and in-solution digestions that identify 88 and 329 proteins from mouse adipose tissue, we identified significantly more proteins [34,35]. Additionally, while the number of reported proteins here is fewer than other literature values, this study analyzes a single tissue type, whereas other groups analyze multiple tissue types and combine the results [8,36]. The data reported in Table 2 confirmed the efficacy of S-Traps for use in the sample preparation of adipose tissue for proteomic analysis. Of interest were proteins unique to each cohort (Figure 3A,B) and proteins differentially expressed among the comparison groups (Figure 3C–H). For example, the generational obese (Wes_(WES)_) cohort contained 86 unique proteins (Figure 3B), of which 20 were membrane-bound proteins.

### 3.3. Protein Expression Changes in Murine Omenta Are Diet-Dependent

To evaluate the potential for diet-induced remodeling of the omental adipose proteome, volcano plots were used to examine the differentially regulated proteins for the four comparison groups from Table 1 (*p* < 0.05, fold change ≥ 1.2) (Figure 3C–F). Differentially regulated proteins for the three comparison groups from Appendix A (*p* < 0.05, fold change ≥ 1.2) are shown in the volcano plots in Appendix A. The generational obesity comparison group contained the greatest number of differentially regulated proteins out of all seven group comparisons, with 23 up-regulated and 43 down-regulated proteins in the generational obese cohort (Wes_(WES)_) compared to the control-only fed mice (Con_(CON)_) (Figure 3C). Several notable up-regulated proteins included leucine-rich alpha-2-glycoprotein 1 (LRG1; Figure 3C,I), apolipoprotein A4 (APOA4; Figure 3C and Figure 5C), and hemopexin (HPX; Figure 3C,J). When comparing the control diet-fed cohorts (maternal influence), there were 7 up-regulated proteins and 24 down-regulated proteins (Figure 3D; Appendix A), whereas offspring from obese dams (obese maternal influence) had 14 up-regulated proteins and 12 down-regulated proteins (Figure 3E; Appendix A). When comparing obese dams to control dams, 21 proteins were up-regulated, and 7 proteins were down-regulated (Figure 3F; Appendix A). The percentage/number of shared down-regulated or up-regulated proteins (*p* < 0.05) among the different groups is shown in Figure 3G,H. The generational obesity comparison group had 31 unique down-regulated proteins and 19 up-regulated proteins, whereas the maternal obesity comparison group had 6 unique down-regulated proteins and 19 up-regulated proteins (Figure 3G,H). Cytokine array analysis of omental adipose determined dietary differences in C-reactive protein (CRP) expression. Unexpectedly, Wes_(CON)_ mice had the lowest CRP expression, followed by Con_(CON)_ mice. Western diet-fed offspring had high CRP expression among the dietary cohorts; however, generational obese mice had the highest CRP expression in omental adipose (Figure 3K). It is noteworthy that the circulating serum CRP expression (Figure 3L) was significantly higher in offspring from Western-diet dams (Wes_(CON)_ and Wes_(WES)_) compared to offspring from control-diet dams (Con_(CON)_ and Con_(WES)_).

### 3.4. Bioinformatic Analysis of Differentially Expressed Proteins Reveals Omental Adipose Functional Differences

To investigate what biological processes, cellular components, and molecular functions were enriched by the diet-induced proteome remodeling, all significantly up- and down-regulated proteins were submitted to the PANTHER Classification System (Figure 4 and Appendix A; Appendix A). PANTHER gene ontology (GO) analysis of biological processes demonstrated that most of the generational obesity up-regulated proteins were in cellular processes (25%) and metabolic processes (26%) and represented a higher percentage of up-regulated proteins in these processes when compared to the other dietary groups (Figure 4A). Additionally, the immune system process (8%) and response to stimulus (10%) had fewer up-regulated proteins in the generational obesity group compared to the other groups. There was a similar percentage (12%) of regulated proteins enriched in biological processes involved in interspecies interaction between organisms and similar percentages (8–11%) of enriched proteins related to localization among the groups.

The cellular component GO enrichment analysis only showed two significantly enriched components: cellular anatomical entity (subcellular localization) and protein-containing complex (Figure 4B). The cellular anatomical entity displayed the most up-regulated proteins across all dietary groups, with slight differences among the groups. Notably, most proteins identified in all groups were localized to the extracellular space (GO:0005615) or the plasma membrane (GO:0005886; Appendix A). The maternal obesity group had the fewest up-regulated proteins (16%) in the protein-containing complex (GO:0032991) compared to all other groups, which had approximately the same percentage of up-regulated proteins (25–28%).

In terms of molecular function enrichment analysis, most of the up-regulated proteins are related to binding, transporter activity, and molecular function regulation (Figure 4C). Interestingly, the generational obesity group contained functionally enriched proteins related to the aforementioned functions as well as ATP-dependent activity, structural molecule activity, translation regulator activity, and immune system response.

### 3.5. Pathway Enrichment Analysis

Canonical pathways regulated by dietary-influenced omental adipose proteins were determined by Ingenuity Pathway Analysis (IPA, Qiagen). The top 10 canonical pathways overrepresented by the differentially regulated proteins between the dietary groups are shown in Figure 5 and Appendix A and Appendix A. Pathways associated with cancer, immune response, metabolic disease, inflammatory disease, hereditary and development disorders, and lipid and nucleic metabolism were differentially regulated in the dietary groups. In general, pathways associated with IL-15 signaling, B-cell receptor signaling, and systemic lupus erythematosus in B-cell signaling were differentially regulated in all the groups. These pathways all had an abundance of up-regulated immunoglobulin proteins.

There were twenty-six canonical pathways unique to the generational obesity group. Among the top ten canonical pathways shown in Figure 5A, the tRNA charging and maturity-onset diabetes of young (MODY) signaling, which both play a role in hereditary disorders, were exclusively overrepresented in the generational obese omenta. Four proteins—methionyl-tRNA synthetase 1 (MARS1), seryl-tRNA synthetase 1 (SARS1), threonyl-tRNA synthetase 1 (TARS1), and valyl-tRNA synthetase 1 (VARS1)—were mapped to the tRNA charging pathway (Figure 5B), and three proteins—aldolase, fructose-biphosphate B (ALDOB); apolipoprotein A4 (APOA4); and apolipoprotein E (APOE)—were mapped to the MODY signaling pathway (Figure 5C). Additionally, the IL-7 signaling pathway, which plays a role in cellular immune response, was exclusively overrepresented in the generational obesity group. Three immunoglobulin proteins were mapped to the IL-7 pathway (Appendix A). The maternal influence group had twenty-three canonical pathways that were unique to this group. Heat shock protein 90 alpha family class A member 1 (HSP90AA1) and heat shock protein 90 alpha family class B member 1 (HSP90AB1) were mapped to twenty-two of the twenty-three overrepresented pathways. From the top ten canonical pathways overrepresented, five pathways were exclusively overrepresented in the maternal influence group and play a role in cancer, immunological disease, cell cycle, and vitamin and mineral metabolism (Figure 5D). The maternal obesity comparison group had twelve canonical pathways overrepresented, of which four were unique to only this group (Figure 5E). These exclusively overrepresented canonical pathways are involved in metabolic pathways, the immune response, endocrine system development, developmental/hereditary disorders, and fatty acid β-oxidation I pathway, which has been considered a cause in the development of obesity and metabolic diseases, including diabetes mellitus [37]. The proteins mapped to the four unique pathways were phenylalanine-4-hydroxylase (PAH), Pterin-4-alpha-carbinolamine dehydratase (PCBD1; Figure 5F), bone marrow proteoglycan (PRG2; Figure 5G), and 3-ketoacyl-CoA thiolase A (ACAA1; Figure 5H). The obese maternal influence group had five unique canonical pathways that are involved in diabetes mellitus, cancer, and metastasis (Figure 5I).

The generational obesity, obese maternal influence, and maternal obesity groups all have overrepresented FXR/RXR and LXR/RXR canonical pathways. These pathways play a role in inflammatory and lipid metabolism. The generational obesity and obese maternal influence groups had three canonical pathways overrepresented in common that are involved in cancer, inflammatory disease, vitamin and mineral metabolism, and tissue development.

**Figure 5 nutrients-16-03086-f005:**
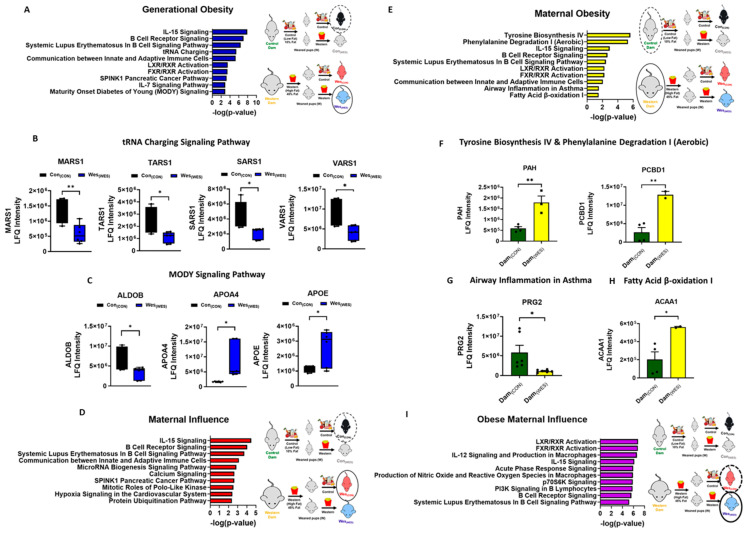
Analysis of differentially expressed proteins. (**A**) Ingenuity pathway analysis of differentially regulated proteins among the generational obesity comparison group. (**B**) Differentially expressed proteins mapped to the tRNA charging signaling pathway. (**C**) Differentially expressed proteins mapped to the MODY signaling pathway. Ingenuity pathway analysis of differentially regulated proteins among the (**D**) maternal influence and (**E**) maternal obesity groups. Differentially expressed proteins mapped to the (**F**) tyrosine biosynthesis and phenylalanine degradation signaling pathway. (**G**) Airway inflammation in asthma signaling pathway and (**H**) fatty acid β-oxidation I signaling pathway. (**I**) Ingenuity pathway analysis of differentially regulated proteins among the obese maternal influence groups. The top 10 pathways are represented in the ingenuity pathway analysis figures. * *p* < 0.05; ** *p* < 0.01.

## 4. Discussion

This study offers a global and unbiased proteomic analysis of visceral omental adipose from dietary cohorts to define molecular features of proteome remodeling that may promote functional changes in female mice. The omentum is a specialized adipose tissue that controls peritoneal homeostasis and is the preferred site of metastasis in several cancers, including gastric, pancreatic, and ovarian cancer [38,39,40,41,42]. Several pre-clinical studies have addressed diet influences on white adipose depots by proteomic analysis [43,44,45,46]; however, these studies pooled various distinct depots together or used male murine samples. Unlike previous studies that have investigated male offspring of obese dams, this study investigates female offspring from both obese and lean dams fed either a control low-fat diet or a Western high-fat diet, resulting in four relevant and distinct cohorts for comparison. Sex differences have been observed between male and female adipose and are potentially due to the X and Y chromosome influences and sex hormone differences, suggesting the importance of investigating both female and male adipose [47,48]. This study is the first to assess female offspring omental adipose from four dietary cohorts to understand the combined impact of maternal and offspring diet on adipose structure and function. Our results support several lines of evidence indicating that a high-fat diet increases visceral adipose mass and adipocyte size [49,50,51]. Adipocyte size is mainly dependent on the triglyceride content [52]. The adipocyte hypertrophy observed in the generational obese cohort supports an increase in triglyceride content and concern regarding the long-term health implications of the generational consumption of high-fat diets. Studies have established a close association between insulin resistance and dyslipidemia with an increase in adipocyte cell size in humans, which highlights the need for effective dietary interventions to mitigate these conditions [49]. Interestingly, while the omental weights of offspring fed a Western diet were similar, there was no corresponding increase in adipocyte size in the transitional obese offspring. This disparity may be attributed to a protective effect resulting from the dam’s diet. Overall, this study provides new insights into the complex relationship between diet and adipose tissue development in female offspring, with important implications for public health.

The use of S-Traps during sample preparation enabled efficient digestion of proteins, including membrane proteins, from the omenta. Using lysate from colorectal cancer cells as the input material, Hummon and co-workers compared the S-Trap sample preparation method to in-solution and FASP workflows and found that the S-Trap method identified the most proteins and peptides [53]. Elinger et al. found that S-Traps are compatible with typical protein extraction buffers and detergents [54]. Typically, membrane proteins are difficult to extract and digest due to their hydrophobic properties, which necessitates high detergent concentrations to mimic the lipid-rich environment of the membrane. Using S-Traps, we identified over 1000 proteins in every dietary cohort and a total of 115 differentially expressed proteins between the groups, of which 64 were down-regulated, and 51 were up-regulated. The use of S-Traps allowed for the identification of 15–73 membrane-bound proteins in each dietary cohort, suggesting S-Traps are superior in adipose protein extraction compared to FASP, which did not detect membrane-bound proteins [51]. PCA analysis indicated the dietary cohorts were readily distinguished. Our study demonstrated the generational obesity comparison group (lean vs. generational obese) had the greatest number of differentially regulated proteins out of all seven group comparisons, suggesting an additive effect passed through generations from an obese mother to her obese offspring. The proteins identified from omental adipose from generational obese offspring were more active in cellular and metabolic processes. Conversely, the number of overexpressed proteins involved in the immune system processes was lower in this group compared to all other comparisons. Importantly, studies have shown that obesity is associated with impaired immune function, with recent findings recognizing adipose tissue as an immune organ, emphasizing a vital role in the functioning of the immune system [55].

LRG1 and HPX were among the twenty-three up-regulated proteins in the generational obesity comparison group. LRG1 is a multi-functional circulating adipokine that is secreted by adipose tissue and promotes lipogenesis [56]. HPX is a glycoprotein that is involved in regulating inflammation and plays a role in adipogenesis and adipocyte differentiation [57,58]. CRP is a nonspecific marker of inflammation and is significantly increased in obese patients. CRP is positively associated with an increased risk of insulin resistance and diabetes in both adolescents and adults [56]. Cytokine array analysis of omental adipose confirmed a significantly elevated level of CRP in generational obese mice compared to the other dietary cohorts. This finding suggests that obesity across generations has a cumulative effect on CRP expression levels, leading to increased inflammation and, consequently, increased risk of developing obesity-associated disorders. Interestingly, high levels of CRP are positively correlated with an increase in LRG1 and HPX, suggesting the potential interplay between these proteins in the context of obesity [56,59]. Suppressing the expression and function of either LRG1 or HPX presents a promising strategy for combating obesity and obesity-related metabolic diseases. This approach could lead to innovative and targeted interventions designed to address the health challenges associated with these conditions.

IL-15 signaling, B-cell receptor signaling, and systemic lupus erythematosus in B-cell signaling were differentially regulated among all the groups. These pathways had an abundance of immunoglobulin proteins that were up-regulated. The production of proinflammatory immunoglobulins has been linked to B-cell-mediated inflammation. It has been suggested that a high-fat diet induces changes in the immunoglobulin heavy chain supply in B cells of adipose tissue [60]. B cells infiltrate visceral adipose, causing functional and phenotypic changes in response to diet-induced obesity. Additionally, B cells govern systemic and local adipose tissue inflammation, as well as obesity-related insulin resistance [61,62]. These findings suggest that B cells play a pivotal role in obesity-induced inflammation and associated diseases.

An unexpected finding from this study was that tRNA charging represents one of the most pronounced differences unique to the generational obesity group. Transfer RNA (tRNA) charging is created by the linkage of an amino acid and a tRNA, resulting in an aminoacyl-tRNA for each amino acid, and it must occur to initiate translation and protein synthesis [63]. Both genetic and environmental effects on tRNAs, causing tRNA aminoacylation, modification, and fragmentation, have emerged as novel contributors to diabetes and obesity. Interestingly, diet-induced tRNA fragments have been linked to the intergenerational inheritance of metabolic traits, indicating that maternal lifestyle choices and habits can have long-lasting effects on the health of offspring [63,64].

Overrepresentation in the MODY signaling pathway was also exclusive to the generational obesity group. MODY is a rare form of familial diabetes mellitus that is usually diagnosed in young adulthood [65,66,67]. Among proteins in the MODY pathway, ALDOB expression was decreased in generational obese mice. ALDOB is an enzyme involved in fructose metabolism that also plays a role in glucogenesis and glycolysis [68]. ALDOB deficiency in humans causes an accumulation of fructose-1-phosphate that subsequently leads to hypoglycemia. Furthermore, ALDOB deficiency has been shown to cause hepatic fat accumulation in humans [69], as well as hereditary fructose intolerance, which ultimately leads to liver failure [70]. The reduction of this enzyme in the Wes_(WES)_ offspring suggests a detrimental consequence of generational obesity.

Both APOA4 and APOE were increased in the generational obese (Wes_(WES)_) cohort compared to the lean (Con_(CON)_) cohort. APOA4 is the most abundant apolipoprotein that facilitates lipid transport and metabolism. APO4A is synthesized in the small intestine and transported by circulation to the adipose, among other tissues. APOA4 is involved in several aspects of glucose homeostasis, including the promotion of glucose uptake in adipocytes [71], and is proposed to be an early diagnostic biomarker of prediabetes [72], impaired renal function [73], and liver fibrosis [74,75]. APOE is a multi-functional protein highly expressed in adipose tissue, where it modulates adipocyte lipid influx. The overexpression of APOE has been shown to predispose mice to diet-induced obesity, hyperglycemia, and insulin resistance, suggesting generational obese mice (Wes_(WES)_) have a high propensity for developing hyperglycemia and insulin resistance [76,77]. Additionally, the overexpression of APOE has also been associated with tumor progression and poor survival, which could play a role in cancers that develop in the peritoneal cavity or metastasize to the adipose-rich omentum [78,79].

PAH, PCBD1, and ACAA1 were increased in high-fat diet-fed dams (Dam_(WES)_) compared to low-fat diet-fed dams (Dam_(CON)_). PCBD1 is an enzyme responsible for recycling tetrahydrobiopterin. Tetrahydrobiopterin works with PAH to convert phenylalanine to tyrosine. A deficiency in the PAH enzyme leads to hyperphenylalaninemia, a recessive inherited metabolic condition [80,81]. Increased circulating concentrations of phenylalanine and tyrosine have both been reported to be increased in the obese and in type 2 diabetes. PRG2 was down-regulated in obese dams compared to lean dams. PRG2 is the predominant component of the eosinophil leukocyte granule. Adipose eosinophils are reduced in obesity [82], and an increase in eosinophils prevented high-fat diet-induced weight gain, suggesting that therapeutic targeting of adipose eosinophils may reduce inflammation and body fat [83]. Additionally, low PRG2 expression has been associated with drug resistance in chronic myeloid leukemia [84]. ACAA1 is a key regulator of fatty acid β-oxidation in peroxisomes and lipid metabolism. Its function is to catalyze the cleavage of 3-ketoacyl-CoA to acetyl-CoA and acyl-CoA, which contribute to the synthesis and degradation of fatty acids [85]. ACAA1 plays a role in positively regulating cell viability, lipogenesis, and triglyceride accumulation [86].

## 5. Conclusions

Maternal obesity is a global public health concern that affects the short- and long-term health of the mother and offspring. The increase in the prevalence of childhood obesity is also a major concern, as the onset of obesity can be influenced by both environmental and genetic factors. Fetal overgrowth in human pregnancy is often the result of maternal obesity and is a risk factor for developing obesity later in life (generational obesity) [87]. Furthermore, maternal obesity may play a direct role in the transmission of obesity-related traits from generation to generation [88]. This study provides an unbiased proteomic analysis of the omental adipose proteome from distinct maternal and offspring dietary cohorts and reveals that generational obesity is a detrimental phenomenon that warrants further investigation. The data collected herein provide a valuable resource and novel insight into the role of generational obesity on the peritoneal cavity, specifically the adipose-rich omentum. Further exploration of the differentially expressed proteins is warranted, as a better understanding of the impact of generational obesity on the omentum will help improve the characterization of obesity and its associated health conditions and will aid in the discovery of therapeutic interventions that will ultimately improve outcomes and quality of life for those affected by these conditions.

## Figures and Tables

**Figure 1 nutrients-16-03086-f001:**
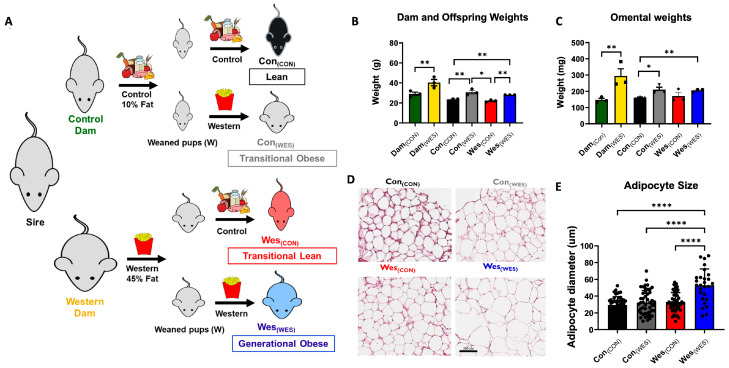
Experimental design used to investigate generational obesity and visceral adipose characteristics. (**A**) Control and Western diet-fed dams were bred with the same sire, and each had a litter of pups. Female offspring from the control-diet dams (Dam_(CON)_) were fed a control diet (10% fat) or a Western diet (45% fat), resulting in two dietary cohorts: Con_(CON)_ and Con_(WES)_, respectively. Offspring from Western-diet dams (Dam_(WES)_) were fed either a control diet (10% fat) or a Western diet (45% fat), resulting in two dietary cohorts: Wes_(CON)_ and Wes_(WES)_, respectively. All mice were sacrificed, and the omentum was excised from each animal. (**B**) Dam and offspring body weight at the time of omenta excision (*n* = 3). (**C**) Omental weights from dietary cohorts at the time of excision. (**D**) Representative H&E staining of visceral adipose from each dietary cohort. (**E**) Adipocyte diameter quantified by Image J. Statistical results are presented as the mean ± SEM. *n* = 3 for each dietary cohort. * *p* < 0.05; ** *p* < 0.01; **** *p* < 0.0001.

**Figure 2 nutrients-16-03086-f002:**
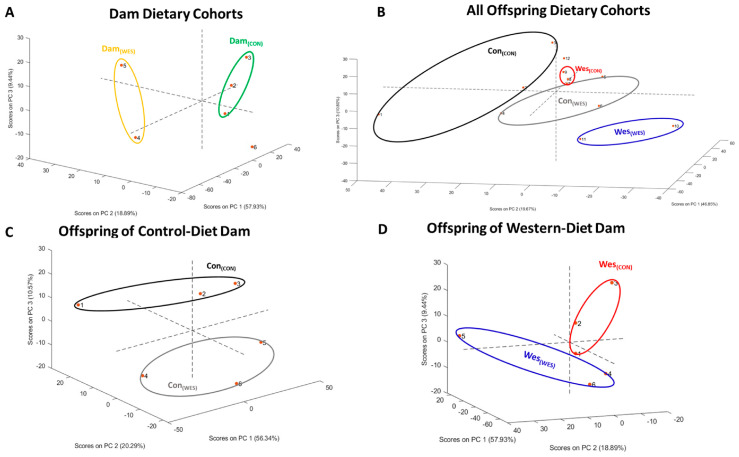
Principal component analysis (PCA) revealed clustering of various dietary cohorts. (**A**) Clustering of dam dietary cohorts. Dots 1–3 represents Dam_(CON)_ and 4–6 represents Dam_(WES)_. (**B**) Clustering of all offspring dietary cohorts. Dots 1–3 represents Con_(CON)_; 4–6 represents Con_(WES)_; 7–9 represents Wes_(CON)_; and 10–12 represents Wes_(WES)_. (**C**) Clustering of offspring from control-diet dam. Dots 1–3 represents Con_(CON)_ and 4–6 represents Con_(WES)_.(**D**) Clustering of offspring from Western-diet dam. Dots 1–3 represents Wes_(CON)_ and 4–6 represents Wes_(WES)_.

**Figure 3 nutrients-16-03086-f003:**
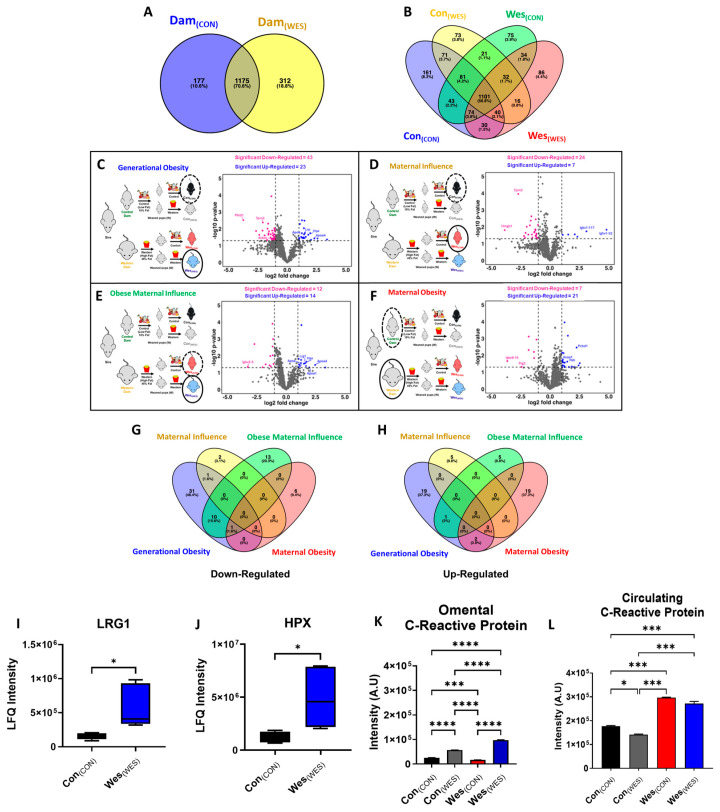
Up- and down-regulation of proteins among dietary cohorts was enabled by proteomic analysis of omental adipose. (**A**,**B**) Venn diagrams comparing omental adipose proteome for (**A**) dam dietary cohorts and (**B**) offspring cohorts. Volcano plots showing up- and down-regulated proteins for (**C**) generational obesity, (**D**) maternal influence, (**E**) obese maternal influence, and (**F**) maternal obesity. Blue dots represent up-regulated proteins, and pink dots represent down-regulated proteins. The study design to the left of each volcano plot shows the two cohorts being compared, with the solid black oval representing the fold change with respect to the cohort circled in black dotted lines. Additional volcano plots of the other groups can be found in Appendix A. Venn diagrams of (**G**) down-regulated and (**H**) up-regulated proteins from comparison groups. Expression of two differentially expressed proteins of interest (**I**) LRG1 and (**J**) HPX between lean and generational obese dietary cohorts. Protein expression of C-reactive protein in (**K**) omental adipose and (**L**) serum from all offspring dietary cohorts. * *p* < 0.05; *** *p* < 0.001; **** *p* < 0.0001.

**Figure 4 nutrients-16-03086-f004:**
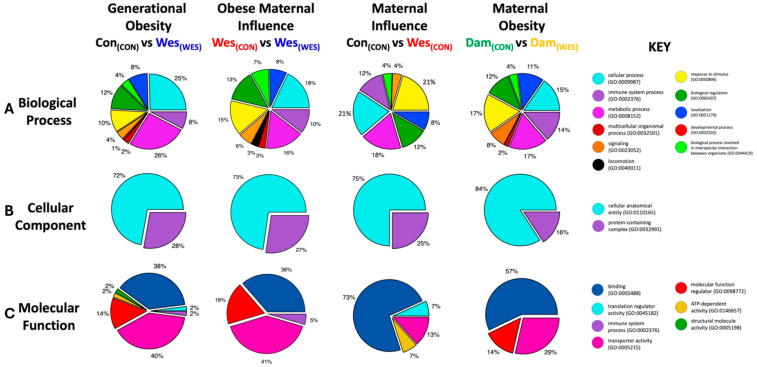
PANTHER gene ontology (GO) analysis identified the processes, location, and function associated with differentially expressed proteins. For each cohort, GO analysis identified the (**A**) biological process, (**B**) cellular component, and (**C**) molecular function of proteins identified by proteomics experiments.

**Table 1 nutrients-16-03086-t001:** Description of cohort comparisons and their given group name. Colors indicate the different dietary cohorts.

Group Name	Cohorts Compared
Generational Obesity	Con_(CON)_ vs. Wes_(WES)_
Obese Maternal Influence	Wes_(CON)_ vs. Wes_(WES)_
Maternal Influence	Con_(CON)_ vs. Wes_(CON)_
Maternal Obesity	Dam_(CON)_ vs. Dam_(WES)_

**Table 2 nutrients-16-03086-t002:** Total number of proteins, peptides, and PSMs identified for each cohort. Values reported are average ± one standard deviation (*n* = 6 injections). Colors indicate the different dietary cohorts.

Cohort	Protein Group IDs	Peptide IDs	MS/MS	PSMs	Percentage of PSMs over MS/MS
Dam_(CON)_	990 ± 31	7230 ± 244	34,158 ± 571	14,471 ± 758	42 ± 2
Dam_(WES)_	1038 ± 110	7424 ± 895	35,133 ± 686	14,848 ± 1666	42 ± 4
Con_(CON)_	1030 ± 45	8468 ± 556	35,648 ± 659	16,733 ± 1246	47 ± 3
Con_(WES)_	1013 ± 69	7342 ± 812	34,798 ± 1134	14,675 ± 1509	42 ± 3
Wes_(CON)_	1139 ± 24	8409 ± 252	35,428 ± 472	16,455 ± 631	46 ± 1
Wes_(WES)_	1045 ± 128	7599 ± 1153	33,994 ± 3072	16,359 ± 3036	48 ± 10

## Data Availability

The mass spectrometry proteomics datasets generated for this study can be found in the ProteomeXchange Consortium via the PRIDE partner repository with the dataset identifier PXD042092.

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
