# Peer review of "Generational Diet-Induced Obesity Remodels the Omental Adipose Proteome in Female Mice"

_nutrients, 2024, doi:10.3390/nu16183086_

Round 1

Reviewer 1 Report

Comments and Suggestions for Authors

The current study used S-trap to enable the label-free quantitative analysis of the adipose proteome in the context of generational obesity and maternal obesity. The database generated will be a valuable resource for the researchers in the field.

1. Although several differentially expressed proteins and pathways have been detected by S-trap, throughout the study, the results are completely descriptive. It would be necessary to validate the major findings using wet experiments, such as immunostaining or western blotting. For example, it is interesting and meaningful to validate that CRP showed differential expression in con (con), con (wes), wes (con), wes (wes) in adipose tissue. The serum CRP level shall be examined as well.

2. Past tense shall be used for description of the result.

3. Wes (Wes) showed larger adipocyte size than Con (Wes), which is interesting. But the fat pad weight was comparable between the two groups. How to explain this observation?

Reviewer 2 Report

Comments and Suggestions for Authors

This manuscript provides a resource for the adipose tissue proteome of female mice fed various diets, offering a variety of meaningful information.

1. Figures in the PDF version are somewhat unclear and difficult to read, so providing all figures in high resolution would be helpful.

2. Figure titles are currently focused mainly on the experimental methods; it would be better to highlight the main conclusions of each Figure.

3. A more meaningful discussion of the roles of the differentially expressed genes identified through proteomic analysis and their potential as targets for obesity therapy is needed.

4. It would be helpful to include several relevant conclusions and add the Conclusion section at the end to clearly describe the important adipose tissue-specific protein factors under different diet compositions.

Round 2

Reviewer 1 Report

Comments and Suggestions for Authors

Generally all the questions have been addressed.